# PseudoSeg: Designing Pseudo Labels for Semantic Segmentation

**Yuliang Zou**[1]* **Zizhao Zhang**[2] **Han Zhang**[3] **Chun-Liang Li**[2]
**Xiao Bian**[2] **Jia-Bin Huang**[1] **Tomas Pfister**[2]
[1]Virginia Tech  [2]Google Cloud AI  [3]Google Brain

## Abstract

Recent advances in semi-supervised learning (SSL) demonstrate that a combination of consistency regularization and pseudo-labeling can effectively improve image classification accuracy in the low-data regime. Compared to classification, semantic segmentation tasks require much more intensive labeling costs. Thus, these tasks greatly benefit from data-efficient training methods. However, structured outputs in segmentation render particular difficulties (e.g., designing pseudo-labeling and augmentation) to apply existing SSL strategies. To address this problem, we present a simple and novel re-design of pseudo-labeling to generate well-calibrated structured pseudo labels for training with unlabeled or weakly-labeled data. Our proposed pseudo-labeling strategy is network structure agnostic to apply in a one-stage consistency training framework. We demonstrate the effectiveness of the proposed pseudo-labeling strategy in both low-data and high-data regimes. Extensive experiments have validated that pseudo labels generated from wisely fusing diverse sources and strong data augmentation are crucial to consistency training for semantic segmentation. The source code is available at https://github.com/googleinterns/wss.

## 1 Introduction

Image semantic segmentation is a core computer vision task that has been studied for decades. Compared with other vision tasks, such as image classification and object detection, human annotation of pixel-accurate segmentation is dramatically more expensive. Given sufficient pixel-level *labeled* training data (i.e., high-data regime), the current state-of-the-art segmentation models (e.g., DeepLabv3+ (Chen et al., 2018)) produce satisfactory segmentation prediction for common practical usage. Recent exploration demonstrates improvement over high-data regime settings with large-scale data, including self-training (Chen et al., 2020a; Zoph et al., 2020) and backbone pre-training (Zhang et al., 2020a).

In contrast to the high-data regime, the performance of segmentation models drop significantly, given very limited pixel-labeled data (i.e., low-data regime). Such ineffectiveness at the low-data regime hinders the applicability of segmentation models. Therefore, instead of improving high-data regime segmentation, our work focuses on data-efficient segmentation training that only relies on few pixel-labeled data and leverages the availability of extra unlabeled or weakly annotated (e.g., image-level) data to improve performance, with the aim of narrowing the gap to the supervised models trained with fully pixel-labeled data.

Our work is inspired by the recent success in semi-supervised learning (SSL) for image classification, demonstrating promising performance given very limited labeled data and a sufficient amount of unlabeled data. Successful examples include MeanTeacher (Tarvainen & Valpola, 2017), UDA (Xie et al., 2019), MixMatch (Berthelot et al., 2019b), FeatMatch (Kuo et al., 2020), and FixMatch (Sohn et al., 2020a). One outstanding idea in this type of SSL is *consistency training*: making predictions consistent among multiple augmented images. FixMatch (Sohn et al., 2020a) shows that using high-confidence one-hot pseudo labels obtained from weakly-augmented unlabeled data to train strongly-augmented counterpart is the key to the success of SSL in image classification.

---

*Work done during internship at Google Cloud AI Research.

However, effective pseudo labels and well-designed data augmentation are non-trivial to satisfy for semantic segmentation. Although we observe that many related works explore the second condition (i.e., augmentation) for image segmentation to enable consistency training framework (French et al., 2020; Ouali et al., 2020), we show that a wise design of pseudo labels for segmentation has great veiled potentials.

In this paper, we propose PseudoSeg, a one-stage training framework to improve image semantic segmentation by leveraging additional data either with image-level labels (weakly-labeled data) or without any labels. PseudoSeg presents a novel design of pseudo-labeling to infer effective structured pseudo labels of additional data. It then optimizes the prediction of strongly-augmented data to match its corresponding pseudo labels. In summary, we make the following contributions:

- We propose a simple one-stage framework to improve semantic segmentation by using a limited amount of pixel-labeled data and sufficient unlabeled data or image-level labeled data. Our framework is simple to apply and therefore network architecture agnostic.
- Directly applying consistency training approaches validated in image classification renders particular challenges in segmentation. We first demonstrate how well-calibrated soft pseudo labels obtained through wise fusion of predictions from diverse sources can greatly improve consistency training for segmentation.
- We conduct extensive experimental studies on the PASCAL VOC 2012 and COCO datasets. Comprehensive analyses are conducted to validate the effectiveness of this method at not only the low-data regime but also the high-data regime. Our experiments study multiple important open questions about transferring SSL advances to segmentation tasks.

## 2 RELATED WORK

**Semi-supervised classification.** Semi-supervised learning (SSL) aims to improve model performance by incorporating a large amount of unlabeled data during training. Consistency regularization and entropy minimization are two common strategies for SSL. The intuition behind consistency-based approaches (Laine & Aila, 2016; Sajjadi et al., 2016; Miyato et al., 2018; Tarvainen & Valpola, 2017) is that, the model output should remain unchanged when the input is perturbed. On the other hand, the entropy minimization strategy (Grandvalet & Bengio, 2005) argues that the unlabeled data can be used to ensured classes are well-separated, which can be achieved by encouraging the model to output low-entropy predictions. Pseudo-labeling (Lee, 2013) is one of the methods for implicit entropy minimization. Recently, holistic approaches (Berthelot et al., 2019b;a; Sohn et al., 2020a) combining both strategies have been proposed and achieved significant improvement. By redesigning the pseudo label, we propose an efficient one-stage semi-supervised learning framework of semantic segmentation for consistency training.

**Semi-supervised semantic segmentation.** Collecting pixel-level annotations for semantic segmentation is costly and prone to error. Hence, leveraging unlabeled data in semantic segmentation is a natural fit. Early methods utilize a GAN-based model either to generate additional training data (Souly et al., 2017) or to learn a discriminator between the prediction and the ground truth mask (Hung et al., 2018; Mittal et al., 2019). Consistency regularization based approaches have also been proposed recently, by enforcing the predictions to be consistent, either from augmented input images (French et al., 2020; Kim et al., 2020), perturbed feature embeddings (Ouali et al., 2020), or different networks (Ke et al., 2020). Recently, Luo & Yang (2020) proposes a dual-branch training network to jointly learn from pixel-accurate and coarse labeled data, achieving good segmentation performance. To push the performance of state of the arts, iterative self-training approaches (Chen et al., 2020a; Zoph et al., 2020; Zhu et al., 2020) have been proposed. These methods usually assume the available labeled data is enough to train a good teacher model, which will be used to generate pseudo labels for the student model. However, this condition might not satisfy in the low-data regime. Our proposed method, on the other hand, realizing the ideas of both consistency regularization and pseudo-labeling in segmentation, consistently improves the supervised baseline in both low-data and high-data regimes.

**Weakly-supervised semantic segmentation.** Instead of supervising network training with accurate pixel-level labels, many prior works exploit weaker forms of annotations (e.g., bounding boxes (Dai et al., 2015), scribbles (Lin et al., 2016), image-level labels). Most recent approaches use image-level labels as the supervisory signal, which exploits the idea of class activation map (CAM) (Zhou et al., 2016). Since the vanilla CAM only focus on the most discriminative region of objects, dif-

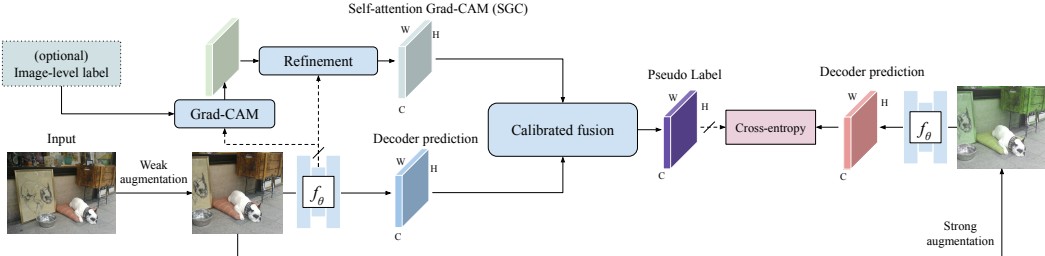

Figure 1: **Overview of unlabeled data training branch.** Given an image, the weakly augmented version is fed into the network to get the decoder prediction and Self-attention Grad-CAM (SGC). The two sources are then combined via a calibrated fusion strategy to form the pseudo label. The network is trained to make its decoder prediction from strongly augmented image to match the pseudo label by a per-pixel cross-entropy loss.

ferent ways to refine CAM have been proposed, including partial image/feature erasing (Hou et al., 2018; Wei et al., 2017; Li et al., 2018), using an additional saliency estimation model (Oh et al., 2017; Huang et al., 2018; Wei et al., 2018), utilizing pixel similarity to propagate the initial score map (Ahn & Kwak, 2018; Wang et al., 2020), or mining and co-segment the same category of objects across images (Sun et al., 2020; Zhang et al., 2020b). While achieving promising results using the approaches mentioned above, most of them require a *multi-stage* training strategy. The refined score maps are optimized again using a dense-CRF model (Krähenbühl & Koltun, 2011), and then used as the target to train a separate segmentation network. On the other hand, we assume there exists a small number of fully-annotated data, which allows us to learn stronger segmentation models than general methods without needing pixel-labeled data.

## 3 THE PROPOSED METHOD

In analogous to SSL for classification, our training objective in PseudoSeg consists of a supervised loss $\mathcal{L}_s$ applied to pixel-level labeled data $\mathcal{D}_l$, and a consistency constraint $\mathcal{L}_u$ applied to unlabeled data $\mathcal{D}_u$ [1]. Specifically, the supervised loss $\mathcal{L}_s$ is the standard pixel-wise cross-entropy loss on the weakly augmented pixel-level labeled examples:

$$\mathcal{L}_s = \frac{1}{N \times |\mathcal{D}_l|} \sum_{x \in \mathcal{D}_l} \sum_{i=0}^{N-1} \text{CrossEntropy}\left(y_i, f_\theta(\omega(x_i))\right), \tag{1}$$

where $\theta$ represents the learnable parameters of the network function $f$ and $N$ denotes the number of valid labeled pixels in an image $x \in \mathbb{R}^{H \times W \times 3}$. $y_i \in \mathbb{R}^C$ is the ground truth label of a pixel $i$ in $H \times W$ dimensions, and $f_\theta(\omega(x_i)) \in \mathbb{R}^C$ is the predicted probability of pixel $i$, where $C$ is the number of classes to predict and $\omega(\cdot)$ denotes the weak (common) data augmentation operations used by Chen et al. (2018).

During training, the proposed PseudoSeg estimates a *pseudo label* $\widetilde{y} \in \mathbb{R}^{H \times W \times C}$ for each strongly-augmented unlabeled data $x$ in $\mathcal{D}_u$, which is then used for computing the cross-entropy loss. The unsupervised objective can then be written as:

$$\mathcal{L}_u = \frac{1}{N \times |\mathcal{D}_u|} \sum_{x \in \mathcal{D}_u} \sum_{i=0}^{N-1} \text{CrossEntropy}\left(\widetilde{y}_i, f_\theta(\beta \circ \omega(x_i))\right), \tag{2}$$

where $\beta(\cdot)$ denotes a stronger data augmentation operation, which will be described in Section 3.2. We illustrate the unlabeled data training branch in Figure 1.

### 3.1 THE DESIGN OF STRUCTURED PSEUDO LABELS

The next important question is how to generate the desirable pseudo label $\widetilde{y}$. A straightforward solution is directly using the decoder output of a trained segmentation model after confidence threshold-

---

[1]For simplicity, here we illustrate the method with unlabeled data and then show it can be easily adapted to use image-level labeled data in Section 3.2.

ing, as suggested by Sohn et al. (2020a); Zoph et al. (2020); Xie et al. (2020); Sohn et al. (2020b). However, as we demonstrate later in the experiments, the generated pseudo hard/soft labels as well as other post-processing of outputs are barely satisfactory in the low-data regime, and thus yield inferior final results. To address this issue, our design of pseudo-labeling has two key insights. First, we seek for a distinct *yet efficient* decision mechanisms to compensate for the potential errors of decoder outputs. Second, wisely fusing multiple sources of predictions to generate an ensemble and better-calibrated version of pseudo labels.

**Starting with localization.** Compared with precise segmentation, learning localization is a simpler task as it only needs to provide coarser-grained outputs than pixel level of objects in images. Based on this motivation, we improve decoder predictions from the localization perspective. Class activation map (CAM) (Zhou et al., 2016) is a popular approach to provide localization for class-specific regions. CAM-based methods (Hou et al., 2018; Wei et al., 2017; Ahn & Kwak, 2018) have been successfully adopted to tackle a different weakly supervised semantic segmentation task from us, where they assume only image-level labels are available. In practice, we adopt a variant of class activation map, Grad-CAM (Selvaraju et al., 2017) in PseudoSeg.

**From localization to segmentation.** CAM estimates the strength of classifier responses on local feature maps. Thus, an inherent limitation of CAM-based approaches is that it is prone to attending only to the most discriminative regions. Although many weakly-supervised segmentation approaches (Ahn & Kwak, 2018; Ahn et al., 2019; Sun et al., 2020) aim at refining CAM localization maps to segmentation masks, most of them have complicated post-processing steps, such as dense CRF (Krähenbühl & Koltun, 2011), which increases the model complexity when used for consistency training. Here we present a computationally efficient yet effective refinement alternative, which is learnable using available pixel-labeled data.

Although CAM only localizes partial regions of interests, if we know the pairwise similarities between regions, we can propagate the CAM scores from the discriminative regions to the rest unattended regions. Actually, it has been shown in many works that the learned high-level deep features are usually good at similarity measurements of visual objects. In this paper, we find hypercolumn (Hariharan et al., 2015) with a learnable similarity measure function works fairly effective.

Given the vanilla Grad-CAM output for all $C$ classes, which can be viewed as a spatially-flatten 2-D vector of weight $m \in \mathbb{R}^{L \times C}$, where each row $m_i$ is the response weight per class for one region $i$. Using a kernel function $\mathcal{K}(\cdot, \cdot) : \mathbb{R}^H \times \mathbb{R}^H \to \mathbb{R}$ that measures element-wise similarity given feature $h \in \mathbb{R}^H$ of two regions, the propagated score $\hat{m}_i \in \mathbb{R}^C$ can be computed as follows

$$\hat{m}_i = \left( m_i + \sum_{j=0}^{L-1} \frac{e^{\mathcal{K}(W_k h_i, W_v h_j)}}{\sum_{k=0}^{L-1} e^{\mathcal{K}(W_k h_i, W_v h_k)}} m_j \right) \cdot W_c. \tag{3}$$

The goal of this function is to train $\Theta = \{W_k, W_v \in \mathbb{R}^{H \times H}, W_c \in \mathbb{R}^{C \times C}\}$ in order to propagate the high value in $m$ to all adjacent elements in the feature space $\mathbb{R}^H$ (i.e., hypercolumn features) to region $i$. Adding $m_i$ in equation 3 indicates the skip-connection. To compute propagated score for all regions, the operations in equation 3 can be efficiently implemented with self-attention dot-product (Vaswani et al., 2017). For brevity, we denote this efficient refinement process output as *self-attention Grad-CAM* (SGC) maps in $\mathbb{R}^{H \times H \times C}$. Figure 6 in Appendix A specifies the architecture.

**Calibrated prediction fusion.** SGC maps are obtained from low-resolution feature maps. It is then resized to the desired output resolution, and thus not sufficient at delineating crisp boundaries. However, compared to the segmentation decoder, SGC is capable of generating more locally-consistent masks. Thus, we propose a novel calibrated fusion strategy to take advantage of both decoder and SCG predictions for better pseudo labels.

Specifically, given a batch of decoder outputs (pre-softmax logits) $\hat{p} = f_\theta(\omega(x))$ and SGC maps $\hat{m}$ computed from weakly-augmented data $\omega(x)$, we generate the pseudo labels $\widetilde{y}$ by

$$\mathcal{F}(\hat{p}, \hat{m}) = \text{Sharpen} \left( \gamma \, \text{Softmax} \left( \frac{\hat{p}}{\text{Norm}(\hat{p}, \hat{m})} \right) + (1 - \gamma) \, \text{Softmax} \left( \frac{\hat{m}}{\text{Norm}(\hat{p}, \hat{m})} \right), T \right). \tag{4}$$

Two critical procedures are proposed to use here to make the fusion process successful. First, $\hat{p}$ and $\hat{m}$ are from different decision mechanisms and they could have very different degrees of overconfidence. Therefore, we introduce the operation $\text{Norm}(a, b) = \sqrt{\sum_i^{|a|} (a_i^2 + b_i^2)}$ as a nor-

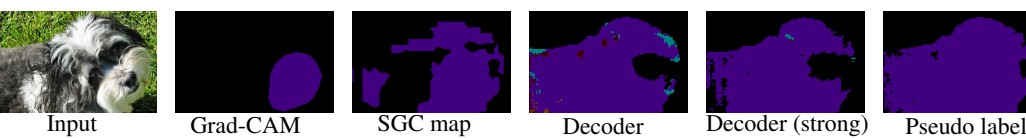

| Input | Grad-CAM | SGC map | Decoder | Decoder (strong) | Pseudo label |

Figure 2: **Visualization of pseudo labels and other predictions.** The generated pseudo label by fusing the predictions from the decoder and SGC map is used to supervise the decoder (strong) predictions of the strongly-augmented counterpart.

malization factor. It alleviates the over-confident probability after softmax, which could unfavorably dominate the resulted $\gamma$-averaged probability. Second, the distribution sharpening operation Sharpen$(a, T)_i = a_i^{1/T} / \sum_j^C a_j^{1/T}$ adjusts the temperature scalar $T$ of categorical distribution (Berthelot et al., 2019b; Chen et al., 2020b). Figure 2 illustrates the predictions from different sources. More importantly, we investigate the pseudo-labeling from a calibration perspective (Section 4.3), demonstrating that the proposed soft pseudo label $\widetilde{y}$ leads to a better calibration metric comparing to other possible fusion alternatives, and justifying why it benefits the final segmentation performance.

**Training.** Our final training objective contains two extra losses: a classification loss $\mathcal{L}_x$, and a segmentation loss $\mathcal{L}_{sa}$. First, to compute Grad-CAM, we add a one-layer classification head after the segmentation backbone and a multi-label classification loss $\mathcal{L}_x$. Second, as specified in Appendix A (Figure 6), SGC maps are scaled as pixel-wise probabilities using one-layer convolution followed by softmax in equation 3. Learning $\Theta$ to predict SGC maps needs pixel-labeled data $D_l$. It is achieved by an extra segmentation loss $\mathcal{L}_{sa}$ between SGC maps of pixel-labeled data and corresponding ground truth. All the loss terms are jointly optimized (i.e., $\mathcal{L}_u + \mathcal{L}_s + \mathcal{L}_x + \mathcal{L}_{sa}$), while $\mathcal{L}_{sa}$ only optimizes $\Theta$ (achieved by stopping gradient). See Figure 7 in the appendix for further details.

## 3.2 INCORPORATING IMAGE-LEVEL LABELS AND AUGMENTATION

The proposed PseudoSeg can easily incorporate image-level label information (if available) into our one-stage training framework, which also leads to consistent improvement as we demonstrate in experiments. We utilize the image-level data with two following steps. First, we directly use ground truth image-level labels to generate Grad-CAMs instead of using classifier outputs. Second, they are used to increase classification supervision beyond pixel-level labels for the classifier head.

For strong data augmentation, we simply follow color jittering operations from SimCLR (Chen et al., 2020b) and remove all geometric transformations. The overall strength of augmentation can be controlled by a scalar (studied in experiments). We also apply once random CutOut (DeVries & Taylor, 2017) with a region of $50 \times 50$ pixels since we find it gives consistent though minor improvement (pixels inside CutOut regions are ignored in computing losses).

## 4 EXPERIMENTAL RESULTS

We start by specifying the experimental details. Then, we evaluate the method in the settings of using pixel-level labeled data and unlabeled data, as well as using pixel-level labeled data and image-level labeled data, respectively. Next, we conduct various ablation studies to justify our design choices. Lastly, we conduct more comparative experiments in specific settings.

To evaluate the proposed method, we conduct the main experiments and ablation studies on the PASCAL VOC 2012 dataset (VOC12) (Everingham et al., 2015), which contains 21 classes including background. The standard VOC12 dataset has 1,449 images as the training set and 1,456 images as the validation set. We randomly subsample 1/2, 1/4, 1/8, and 1/16 of images in the standard training set to construct the pixel-level labeled data. The remaining images in the standard training set, together with the images in the augmented set (Hariharan et al., 2011) (around 9k images), are used as unlabeled or image-level labeled data. To further verify the effectiveness of the proposed method, we also conduct experiments on the COCO dataset (Lin et al., 2014). The COCO dataset has 118,287 images as the training set, and 5,000 images as the validation set. We evaluate on the 80 foreground classes and the background, as in the object detection task. As the COCO dataset is larger than VOC12, we randomly subsample smaller ratios, 1/32, 1/64, 1/128, 1/256, 1/512, of images from the training set to construct the pixel-level labeled data. The remaining images in the

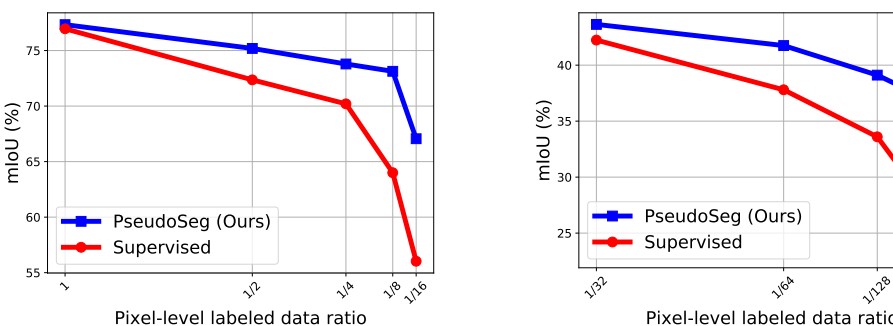

Figure 3: Improvement over the strong supervised baseline, in a semi-supervised setting (w/ unlabeled data) on VOC12 val (left) and COCO val (right).

training set are used as unlabeled data or image-level labeled data. We evaluate the performance using the standard mean intersection-over-union (mIoU) metric. Implementation details can be found in Appendix B.

## 4.1 EXPERIMENTS USING PIXEL-LEVEL LABELED DATA AND UNLABELED DATA

**Improvement over a strong baseline.**   We first demonstrate the effectiveness of the proposed method by comparing it with the DeepLabv3+ model trained with only the pixel-level labeled data. As shown in Figure 3 (a), the proposed method consistently outperforms the supervised training baseline on VOC12, by utilizing the pixel-level labeled data and the unlabeled data. The proposed method not only achieves a large performance boost in the low-data regime (when only 6.25% pixel-level labels available), but also improves the performance when the entire training set (1.4k images) is available. In Figure 3 (b), we again observe consistent improvement on the COCO dataset.

**Comparisons with the others.**  Next, we compare the proposed method with recent state of the arts on both the public 1.4k/9k split (in Table 1) and the created low-data splits (in Table 2), on VOC12. Our method compares favorably with the others.

Table 1: **Comparison with state of the arts on VOC12 val set (w/ pixel-level labeled data and unlabeled data).** We use the official training set (1.4k) as labeled data, and the augmented set (9k) as unlabeled data.

| Method | Network | mIoU (%) |
|---|---|---|
| GANSeg (Souly et al., 2017) | VGG16 | 64.10 |
| AdvSemSeg (Hung et al., 2018) | ResNet-101 | 68.40 |
| CCT (Ouali et al., 2020) | ResNet-50 | 69.40 |
| PseudoSeg (Ours) | ResNet-50 | 71.00 |
| PseudoSeg (Ours) | ResNet-101 | **73.23** |

Table 2: **Comparison with state of the arts on VOC12 val set (w/ pixel-level labeled data and unlabeled data) using low-data splits.** The exact numbers of pixel-labeled images are shown in brackets. All the methods use ResNet-101 as backbone except CCT (Ouali et al., 2020), which uses ResNet-50. * indicates implementation from Ke et al. (2020), ** indicates implementation from French et al. (2020).

| Method | 1/2 (732) | 1/4 (366) | 1/8 (183) | 1/16 (92) |
|---|---|---|---|---|
| AdvSemSeg (Hung et al., 2018) | 65.27 | 59.97 | 47.58 | 39.69 |
| CCT (Ouali et al., 2020) | 62.10 | 58.80 | 47.60 | 33.10 |
| *MT (Tarvainen & Valpola, 2017) | 69.16 | 63.01 | 55.81 | 48.70 |
| GCT (Ke et al., 2020) | 70.67 | 64.71 | 54.98 | 46.04 |
| **VAT (Miyato et al., 2018) | 63.34 | 56.88 | 49.35 | 36.92 |
| CutMix (French et al., 2020) | 69.84 | 68.36 | 63.20 | 55.58 |
| PseudoSeg (Ours) | **72.41** | **69.14** | **65.50** | **57.60** |

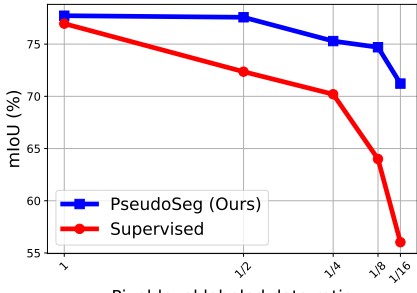 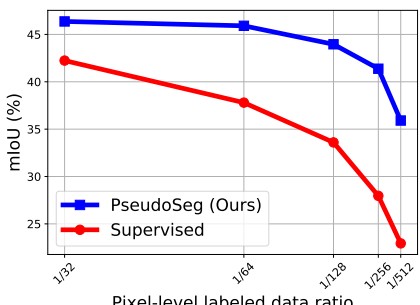

Figure 4: Improvement over the strong supervised baseline, in a semi-supervised setting (w/ image-level labeled data) on VOC12 val (left) and COCO val (right).

Table 3: **Comparison with state of the arts on VOC12 val set (w/ pixel-level labeled data and image-level labeled data).** We use the official training set (1.4k) as labeled data, and the augmented set (9k) as image-level labeled data.

| Method | Model | Network | mIoU (%) |
|---|---|---|---|
| WSSN (Papandreou et al., 2015) | DeepLab-CRF | VGG16 | 64.60 |
| GAIN (Li et al., 2018) | DeepLab-CRF-LFOV | VGG16 | 60.50 |
| MDC (Wei et al., 2018) | DeepLab-CRF-LFOV | VGG16 | 65.70 |
| DSRG (Huang et al., 2018) | DeepLabv2 | VGG16 | 64.30 |
| GANSeg (Souly et al., 2017) | FCN | VGG16 | 65.80 |
| FickleNet (Lee et al., 2019) | DeepLabv2 | ResNet-101 | 65.80 |
| CCT (Ouali et al., 2020) | PSP-Net | ResNet-50 | 73.20 |
| PseudoSeg (Ours) | DeepLabv3+ | ResNet-50 | **73.80** |

Table 4: **Comparison with state of the arts on VOC12 val set with pixel-level labeled data and image-level labeled data.** Four ratios of pixel-level labeled examples are tested. Both CCT (Ouali et al., 2020) and our method use ResNet-50 as backbone.

| Split | CCT | PseudoSeg |
|---|---|---|
| 1/2 | 66.80 | **73.51** |
| 1/4 | 67.60 | **71.79** |
| 1/8 | 62.50 | **69.15** |
| 1/16 | 51.80 | **65.44** |

## 4.2 EXPERIMENTS USING PIXEL-LEVEL LABELED DATA AND IMAGE-LEVEL LABELED DATA

Similar to semi-supervised learning using pixel-level labeled data and unlabeled data, we first demonstrate the efficacy of our method by comparing it with a strong supervised baseline. As shown in Figure 4, the proposed method consistently improves the strong baseline on both datasets. In Table 3, we evaluate on the public 1.4k/9k split. The proposed method compares favorably with the other methods. Moreover, we further compare to best compared CCT on the created low-data splits (in Table 4). Both experiments show that the proposed PseudoSeg is more robust than the compared method given less data. On all splits on both datasets, using pixel-level labeled data and image-labeled data shows higher mIoU than the setting using pixel-level labeled data and unlabeled data.

## 4.3 ABLATION STUDY

In this section, we conduct extensive ablation experiments on VOC12 to validate our design choices.

**How to construct pseudo label?** We investigate the effectiveness of the proposed pseudo labeling. Table 5 demonstrates quantitative results, indicating that using either decoder output or SGC alone gives an inferior performance. Naively using decoder output as pseudo labels can hardly work well. The proposed fusion consistently performs better, either with or without additional image-level labels. To further answer why our pseudo labels are effective, we study from the model calibration perspective. We measure the expected calibration error (ECE) (Guo et al., 2017) scores of all the intermediate steps and other fusion variants. As shown in Figure 5 (a), the proposed fusion strategy (denoted as G in the figure) achieves the lowest ECE scores, indicating that the significance of jointly using normalization with sharpening (see equation 4) compared with other fusion alternatives. We hypothesize using well-calibrated soft labels makes model training less affected by label noises. The comprehensive calibration study is left as a future exploration direction.

**Using hypercolumn feature or not?** In Figure 5 (b), we study the effectiveness of using hypercolumn features instead of the last feature maps in equation 3. We conduct the experiments on the 1/16 split of VOC12. As we can see, hypercolumn features substantially improve performance.

**Soft or hard pseudo label?** How to utilize predictions as pseudo labels remains an active question in SSL. Next, we study whether we should use soft or hard one-hot pseudo labels. We conduct

Table 5: **Comparison to alternative pseudo labeling strategies.** We conduct experiments using 1/4, 1/8, 1/16 of the pixel-level labeled data, the exact numbers of images are shown in the brackets.

| Source | Using image-level labels | 1/4 (366) | 1/8 (183) | 1/16 (92) |
|---|---|---|---|---|
| Decoder only | - | 70.22 | 69.35 | 53.20 |
| SGC only | - | 67.07 | 62.61 | 53.42 |
| Calibrated fusion | - | **73.79** | **73.13** | **67.06** |
| Decoder only | ✓ | 73.95 | 73.05 | 67.54 |
| SGC only | ✓ | 71.73 | 67.57 | 64.26 |
| Calibrated fusion | ✓ | **75.29** | **74.70** | **71.22** |

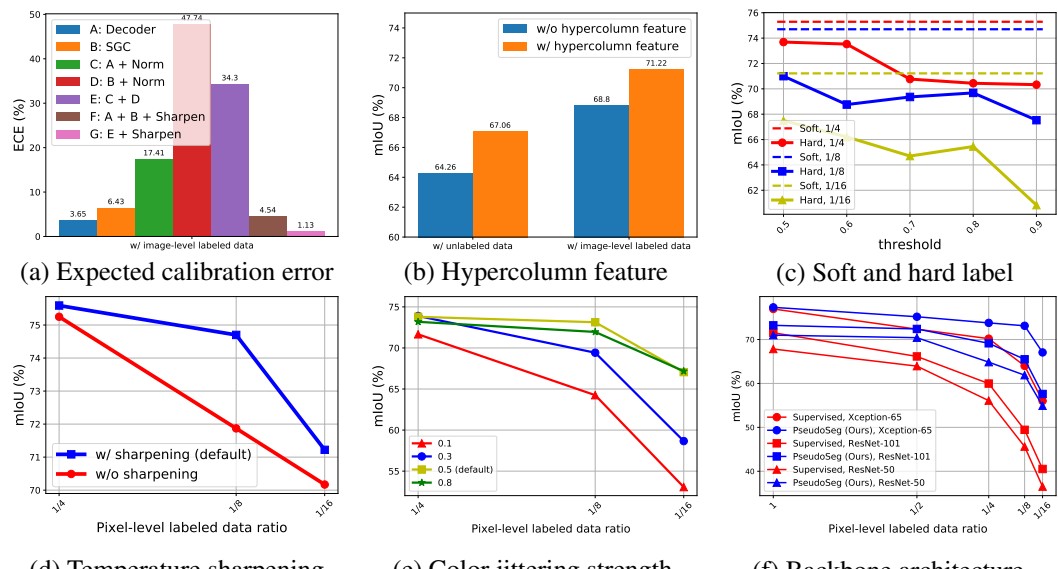

(a) Expected calibration error   (b) Hypercolumn feature   (c) Soft and hard label

(d) Temperature sharpening   (e) Color jittering strength   (f) Backbone architecture

Figure 5: **Ablation studies on different factors.** See Section 4.3 for complete details.

the experiments in the setting where pixel-level labeled data and image-level labeled data are available. As shown in Figure 5 (c), using all predictions as soft pseudo label yields better performance than selecting confident predictions. This suggests that well-calibrated soft pseudo labels might be important in segmentation than over-simplified confidence thresholding.

**Temperature sharpening or not?** We study the effect of temperature sharpening in equation 4. We conduct the experiments in the setting where pixel-level labeled data and image-level labeled data are available. As shown in Figure 5 (d), temperature sharpening shows consistent and clear improvements.

**Strong augmentation strength.** In Figure 5 (e), we study the effects of color jittering in the strong augmentation. The magnitude of jittering strength is controlled by a scalar (Chen et al., 2020b). We conduct the experiments in the setting where pixel-level labeled data and unlabeled data are available. If the magnitude is too small, performance drops significantly, suggesting the importance of strong augmentation.

**Impact of different feature backbones.** In Figure 5 (f), we compare the performance of using ResNet-50, ResNet-101, and Xception-65 as backbone architectures, respectively. We conduct the experiments in the setting where pixel-level labeled data and unlabeled data are available. As we can see, the proposed method consistently improves the baseline by a substantial margin across different backbone architectures.

### 4.4 COMPARISON WITH SELF-TRAINING

Several recent approaches (Chen et al., 2020a; Zoph et al., 2020) exploit the Student-Teacher self-training idea to improve the performance with additional unlabeled data. However, these methods only apply self-training in the high-data regime (i.e., sufficient pixel-labeled data to train teachers).

Table 6: **Comparison with self-training.** We use our supervised baseline as the teacher to generate one-hot pseudo labels, following Zoph et al. (2020).

| Method | Using image-level labels | 1/4 (366) | 1/8 (183) | 1/16 (92) |
|---|---|---|---|---|
| Supervised (Teacher) | - | 70.20 | 64.00 | 56.03 |
| Self-training (Student) | - | 72.85 | 69.88 | 64.20 |
| PseudoSeg (Ours) | - | 73.79 | 73.13 | 67.06 |
| PseudoSeg (Ours) | ✓ | **75.29** | **74.70** | **71.22** |

Here we compare these methods in the low-data regimes, where we focus on. To generate offline pseudo labels, we closely follow segmentation experiments in Zoph et al. (2020): pixels with a confidence score higher than 0.5 will be used as one-hot pseudo labels, while the remaining are treated as ignored regions. This step is considered important to suppress noisy labels. A student model is then trained using the combination of unlabeled data in VOC12 train and augmented sets with generated one-hot pseudo labels and all the available pixel-level labeled data. As shown in Table 6, although the self-training pretty well improves over the supervised baseline, it is inferior to the proposed method [2]. We conjecture that the teacher model usually produces low confidence scores to pixels around boundaries, so pseudo labels of these pixels are filtered in student training. However, boundary pixels are important for improving the performance of segmentation (Kirillov et al., 2020). On the other hand, the design of our method (online soft pseudo labeling process) bypass this challenge. We will conduct more verification of this hypothesis in future work.

### 4.5 IMPROVING THE FULLY-SUPERVISED METHOD WITH ADDITIONAL DATA

We have validated the effectiveness of the proposed method in the low-data regime. In this section, we want to explore whether the proposed method can further improve supervised training in the full training set using additional data. We use the training set (1.4k) in VOC12 as the pixel-level labeled data. The additional data contains additional VOC 9k ($V_{9k}$), COCO training set ($C_{tr}$), and COCO unlabeled data ($C_u$). More training details can be found in Appendix D. As shown in Table 7, the proposed PseudoSeg is able to improve upon the supervised baseline even in the high-data regime, using additional unlabeled or image-level labeled data.

Table 7: **Improving fully supervised model with extra data.** No test-time augmentation is used.

| Method | Baseline | PseudoSeg (w/o image-level labels) | | PseudoSeg (w/ image-level labels) | |
|---|---|---|---|---|---|
| Extra data | - | $C_{tr}+C_u$ | $C_{tr} + C_u + V_{9k}$ | $C_{tr}$ | $C_{tr} + V_{9k}$ |
| mIoU (%) | 76.96 | 77.40 (+0.44) | 78.20 (+1.24) | 77.80 (+0.84) | 79.28 (+2.32) |

## 5 DISCUSSION AND CONCLUSION

The key to the good performance of our method in the low-data regime is the novel re-design of pseudo-labeling strategy, which pursues a different decision mechanism from weakly-supervised localization to "remedy" weak predictions from segmentation head. Then augmentation consistency training progressively improves segmentation head quality. For the first time, we demonstrate that, with well-calibrated soft pseudo labels, utilizing unlabeled or image-labeled data significantly improves segmentation at low-data regimes. Further exploration of fusing stronger and better-calibrated pseudo labels worth more study as future directions (e.g., multi-scaling). Although color jittering works within our method as strong data augmentation, we have extensively explored geometric augmentations (leveraging STN (Jaderberg et al., 2015) to align pixels in pseudo labels and strongly-augmented predictions) for segmentation but find it not helpful. We believe data augmentation needs re-thinking beyond current success in classification for segmentation usage.

### ACKNOWLEDGEMENT

We thank Liang-Chieh Chen and Barret Zoph for their valuable comments.

---

[2]It is difficult to directly compare to Zoph et al. (2020) in their setting because of enormous parallel training, uncommon backbones, and inaccessible pre-training datasets.

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

APPENDIX

## A SELF-ATTENTION GRAD-CAM

We elaborate the detailed pipeline of generating Self-attention Grad-CAM (SGC) maps (equation 3) in Figure 6. To construct the hypercolumn feature, we extract the feature maps from the last two convolutional stages of the backbone network and concatenate them together. We then project the hypercolumn feature to two separate low-dimension embedding spaces to construct "key" and "query", using two $1 \times 1$ convolutional layers. An attention matrix can then be computed via matrix multiplication of "key" and "query". To construct "value", we compute Grad-CAM for each foreground class and then concatenate them together. This results in a $H \times W \times (C - 1)$ score map, where the maximum score of each category is normalized to one separately. We then use image-level labels (either from classifier prediction or ground truth annotation) to set the score maps of non-existing classes to be zero. For each pixel localization, we use one to subtract the maximum score to construct the background score map, which is then concatenated with the foreground score maps to form "value" ($H \times W \times C$). The attention score matrix can then be used to reweight and propagate the scores in "value". The propagated score is added back to the "value" score map, and the pass through a $1 \times 1$ convolution (w/ batch normalization) to output the SGC map.

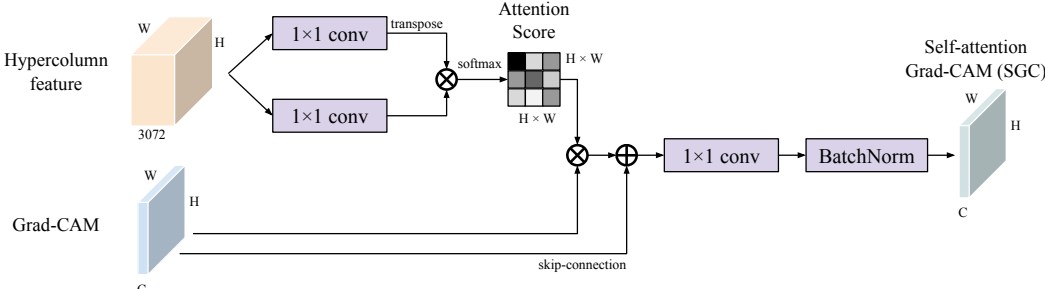

Figure 6: **Diagram of Self-attention Grad-CAM (SGC) .**

## B IMPLEMENTATION DETAILS

We implement our method on top of the publicly available official DeepLab codebase.[3] Unless specified, we adopt the DeepLabv3+ model with Xception-65 (Chollet, 2017) as the feature backbone, which is pre-trained on the ImageNet dataset (Russakovsky et al., 2015). We train our model following the default hyper-parameters (e.g., an initial learning rate of 0.007 with a polynomial learning rate decay schedule, a crop size of $513 \times 513$, and an encoder output stride of 16), using 16 GPUs [4]. We use a batch size of 4 for each GPU for pixel-level labeled data, and 4 for unlabeled/image-level labeled data. For VOC12, we train the model for 30,000 iterations. For COCO, we train the model for 200,000 iterations. We set $\gamma = 0.5$ and $T = 0.5$ unless specified. We do not apply any test time augmentations.

## C LOW-DATA SAMPLING IN PASCAL VOC 2012

Unlike random sampling in image classification, it is difficult to sample uniformly in a low-data case for semantic segmentation due to the imbalance of rare classes. To avoid the missing classes at extremely low data regimes, we repeat the random sampling process for 1/16 three times (while ensuring each class has a certain amount) and report the results. We use Split 1 in the main manuscript. All splits will be released to encourage reproducibility. The results of all the three splits are shown as in Table 8.

---

[3]https://github.com/tensorflow/models/tree/master/research/deeplab

[4]We do not adopt synchronous batch normalization, which is known can improve performance generally.

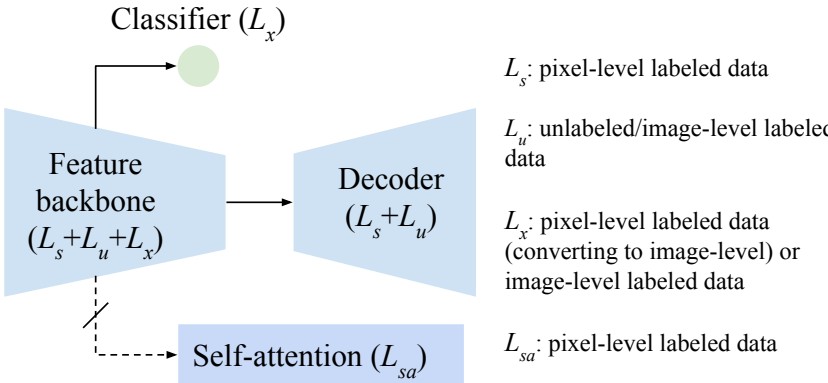

Classifier ($L_x$)

$L_s$: pixel-level labeled data

$L_u$: unlabeled/image-level labeled data

$L_x$: pixel-level labeled data (converting to image-level) or image-level labeled data

$L_{sa}$: pixel-level labeled data

Figure 7: **Training.** For each network component, we show the loss supervision and the corresponding data.

Table 8: **Full results of 1/16 split in VOC12.**

| Method | Using image-level labels | Split 1 | Split 2 | Split 3 |
|---|---|---|---|---|
| Supervised | - | 56.03 | 56.87 | 55.92 |
| PseudoSeg (Ours) | - | 67.06 | 64.12 | 66.09 |
| PseudoSeg (Ours) | ✓ | 71.22 | 68.11 | 69.72 |

## D  HIGH-DATA EXPERIMENTAL SETTINGS

Here we provide more details about the experiments in Section 4.5. Since we have a lot more unlabeled/image-level labeled data, we adopt a longer training schedule (90,000 iterations) [5]. We also adopt a slightly different fusion strategy in this setting by using $T = 0.7$ and $\gamma = 0.3$.

## E  COMPARISON WITH WEAKLY-SUPERVISED APPROACHES

In Table 9, we benchmark recent weakly supervised semantic segmentation performance on PAS-CAL VOC 2012 val set. Instead of enforcing the consistency between different augmented images as we do, these approaches tackle the semantic segmentation task from a different perspective, by exploiting the weaker annotations (image-level labels). As we can see, by exploiting the image-level labels with careful designs, weakly-supervised semantic segmentation methods could achieve reasonably well performance. We believe that both perspectives are feasible and promising for low-data regime semantic segmentation tasks, and complementary to each other. Therefore, these designs could be potentially integrated into our framework to generate better pseudo labels, which leads to improved performance.

Table 9: **Benchmarking state-of-the-art weakly supervised semantic segmentation methods.** All the methods use image-level labels from VOC12 training (1.4k) and augmented (9k) sets.

| Method | Pixel-level labeled data | mIoU (%) |
|---|---|---|
| FickleNet (Lee et al., 2019) | - | 64.9 |
| IRNet (Ahn et al., 2019) | - | 63.5 |
| OAA+ (Jiang et al., 2019) | - | 65.2 |
| SEAM (Wang et al., 2020) | - | 64.5 |
| MCIS (Sun et al., 2020) | - | 66.2 |
| PseudoSeg (Ours) | 1/16 (92) | 71.22 |

---

[5]Note that a longer training schedule does not improve the supervised baseline.

## F  PERFORMANCE ANALYSIS FOR TEMPERATURE SHARPENING

We conduct an additional performance analysis for temporal sharpening. We conduct experiments over T on the 1/16 split of VOC using pixel-level labeled data and image-level labeled data. As shown in Table 10, adopting a $T < 1$ for distribution sharpening generally leads to improved performance.

Table 10: **Performance analysis over T.**

| Temperature (T) | mIoU (%) |
|---|---|
| 0.1 | 71.11 |
| 0.3 | 70.11 |
| 0.5 (default) | 71.22 |
| 0.7 | 72.37 |
| 1.0 (no sharpening) | 68.15 |

## G  EXPERIMENTS ON CITYSCAPES

In this section, we conduct additional experiments on the Cityscapes dataset (Cordts et al., 2016). The Cityscapes dataset contains 50 real-world driving sequences. Among these video sequences, 2,975 frames are selected as the training set, and 500 frames are selected as the validation set. Following previous common practice, we evaluate on 19 semantic classes.

**Comparison with state of the art.** We compare our method with the current state-of-the-art method (French et al., 2020), in the setting of using pixel-level labeled and unlabeled data. We randomly subsample 1/4, 1/8, and 1/30 of the training set to construct the pixel-level labeled data, using the first random seed provided by French et al. (2020). Both French et al. (2020) and our method use ResNet-101 as the feature backbone and DeepLabv3+ (Chen et al., 2018) as the segmentation model. As shown in Table 11, the proposed method achieves promising results on all the three label ratios.

Table 11: **Experiments on Cityscapes (w/ pixel-level labeled data and unlabeled data).**

| Method | 1/4 (744) | 1/8 (372) | 1/30 (100) |
|---|---|---|---|
| CutMix (French et al., 2020) | 68.33 | 65.82 | 55.71 |
| PseudoSeg (Ours) | 72.36 | 69.81 | 60.96 |

**Per-class performance analysis.** Next, we provide per-class performance break down analysis. We compare our method with the supervised baseline on the 1/30 split, using pixel-level labeled data and unlabeled data. As shown in Table 12, the distribution of the labeled pixels is severely imbalanced. Although our method does not in particular address the data imbalance issue, our method improves upon the supervised baseline on most of the classes (except for "Wall" and "Pole").

Table 12: **Per-class performance analysis on Cityscapes (w/ pixel-level labeled data and unlabeled data).**

| Class | Road | Sidewalk | Building | Wall | Fence | Pole | Traffic light | Traffic sign | Vegetation | Terrain |
|---|---|---|---|---|---|---|---|---|---|---|
| Pixel ratio (%) | 36.36 | 5.61 | 20.99 | 0.53 | 0.98 | 1.19 | 0.14 | 0.51 | 19.61 | 1.29 |
| Supervised | 96.03 | 71.26 | 87.53 | **19.75** | 29.11 | **52.19** | 50.19 | 68.09 | 89.93 | 45.79 |
| PseudoSeg (Ours) | **96.64** | **75.06** | **88.63** | 19.67 | **34.09** | 51.75 | **58.19** | **69.95** | **90.43** | **50.48** |

| Class | Sky | Person | Rider | Car | Truck | Bus | Train | Motorcycle | Bicycle | |
|---|---|---|---|---|---|---|---|---|---|---|
| Pixel ratio (%) | 3.70 | 1.10 | 0.16 | 6.49 | 0.38 | 0.13 | 0.23 | 0.06 | 0.54 | |
| Supervised | 91.01 | 74.12 | 43.91 | 89.91 | 7.68 | 14.19 | 17.78 | 25.86 | 69.88 | |
| PseudoSeg (Ours) | **92.99** | **75.16** | **46.09** | **91.60** | **20.39** | **26.30** | **22.13** | **43.96** | **71.30** | |

**Discussion.** Although the scene layouts are quite similar for all the full images, it is still feasible to generate different image-level labels through a more aggressive geometric data augmentation (e.g., scaling, cropping, translation, etc.). In practice, standard segmentation preprocessing steps only crop a sub-region of the whole training images. It only contains partial images with a certain subset of image labels, making the training batches have diverse image-level labels (converted from pixel-level labels, in the fully-labeled+unlabeled setting). Moreover, in the fully-labeled+weakly-labeled

setting, in practice, we can collect diverse Internet images and weakly label them, instead of weakly labeling images from Cityscapes.

## H    QUALITATIVE RESULTS

We visualize several model prediction results for PASCAL VOC 2012 (Figure 8) and COCO (Figure 9). As we can see, the supervised baseline struggles to segment some of the categories and small objects, when trained in the low-data regime. On the other hand, PseudoSeg utilizes unlabeled or weakly-labeled data to generate more satisfying predictions.

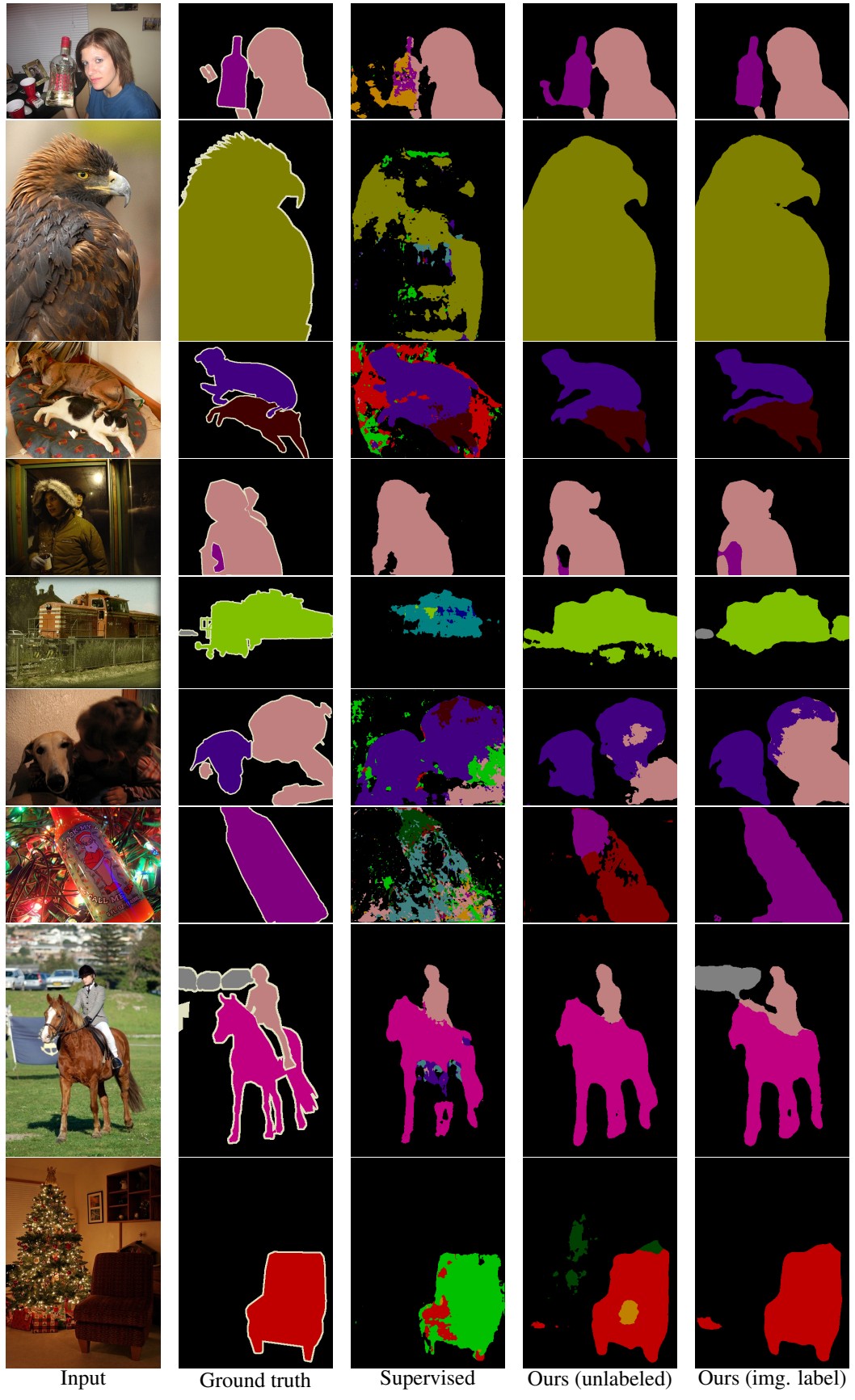

Input   Ground truth   Supervised   Ours (unlabeled)   Ours (img. label)

Figure 8: **Qualitative results of PASCAL VOC 2012.** Models are trained with 1/16 pixel-level labeled data in the training set.

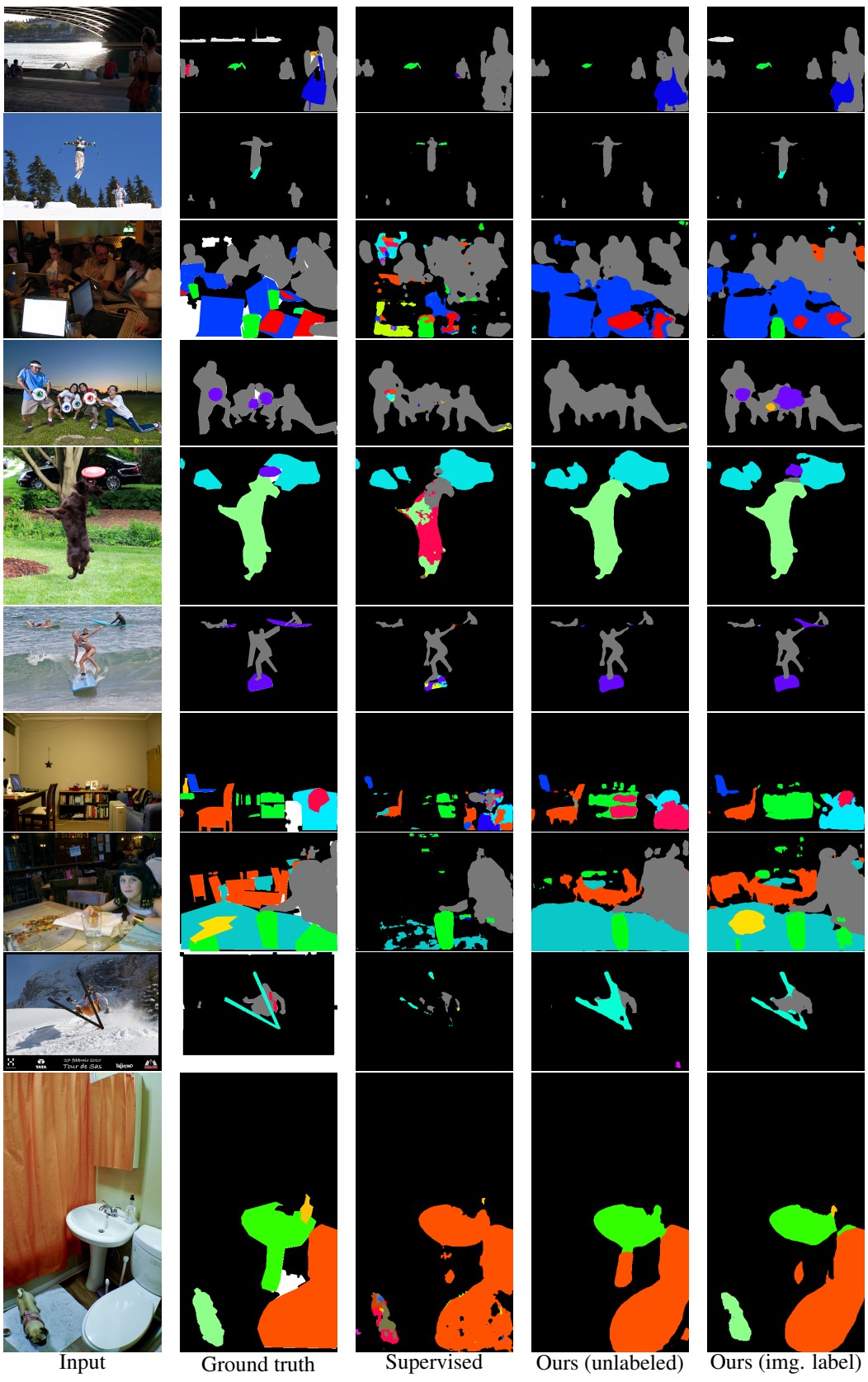

Figure 9: **Qualitative results of COCO.** Models are trained with 1/512 pixel-level labeled data in the training set. Note that white pixel in the ground truth indicates this pixel is not annotated for evaluation.

