# OpenReview forum: "PseudoSeg: Designing Pseudo Labels for Semantic Segmentation"
_ICLR.cc/2021/Conference — ICLR 2021 Poster_

### Official Review · AnonReviewer3 · 2020-10-28
**Interesting paper, good results**

**Rating:** 7
**Confidence:** 4

**Review:**

** Summary
This work addresses the task of semi-supervised learning (SSL) in semantic segmentation. Following recent SOTAs in SSL, this work also advocates for the use of pseudo-labels on unlabeled data and heavy data augmentation. The main novelty of this work is the novel way to construct higher-quality pseudo-labels: besides the pixel-wise classifier's probabilistic outputs, the authors leverage as well CAM-based activation maps, named as SGC, as an additional pseudo-label source.  The final set of pseudo-labels is determined by linear combining the two soft pseudo-label sources with temperature adjustment. The authors conducted extensive experiments with lots of ablation studies to validate the proposed framework.

** Strengths:
- The paper is well-written, easy to follow
- Extensive experiments with adequate discussions
- Improvements over SOTAs on the addressed benchmarks.

** Weakness/concerns:
 - Does "distribution sharpen operation" always use temperature < 1? If yes, what is the reason?
 - How is the temperature $T$ is chosen? May the authors produce a performance analysis over T?
 - In Sec 3.4, it's not clear to me the advantage of proposed method on boundaries. CAM-based activations mostly focus on most discriminative areas (usually inner areas). So hardly SGC can find pseudo-labels on boundaries. Why does the proposed method have an advantage there?
 - More and more segmentation works report results in urban datasets like cityscapes or camvid. It would be interesting to see results on those datasets. One interesting aspect in urban datasets is the natural long-tail class distributions, which severely damages performance on minor classes, especially in low-data regime.

---

> ### Author Response · Authors · 2020-11-23
> **Response to R3 - 1**
>
> We thank R3 for commenting on our work as “interesting”, “well-written and easy to follow”, and also providing instructive questions and suggestions.
>
> **- Distribution sharpening.** To sharpen the distribution, we use T < 1 to make the softmax distribution more "peaky". The extreme case is to set T = 0 so that we get a one-hot label vector. Please refer to Sec 3.1 “Calibrated prediction fusion.” or Eq. 7 in MixMatch [A1] for the mathematical definition of distribution sharpening operation. The sharpened pseudo label helps minimize the entropy of distribution for SSL [A2] and is usually considered important for discriminative learning. It has been shown used for many previous works, such as MixMatch [A1] and UDA [A3]. After the normalization and weighted combination in Eq. 4 of our paper, we get a softer probability distribution. We thus use distribution sharpening to get sharper pseudo labels. We conduct an additional performance analysis over T on the pixel-level+image-level labels setting on the 1/16 split of VOC as follows. Note that only T=0.5 is used in all our experiments in the main paper.
>
> | T        | 0.1   | 0.3   | 0.5 (default) | 0.7   | 1.0 (no sharpening) |
> |----------|-------|-------|---------------|-------|---------------------|
> | mIoU (%) | 71.11 | 70.11 | 71.22         | 72.37 | 68.15               |
>
> **- Advantage over self-training on boundaries?** We fully agree with the review, and that is one of our motivations for fusing two sources of predictions. SGC alone indeed cannot generate high-quality pseudo labels around the object boundaries. In comparison to SGC, we empirically observe that the decoder output is better at capturing boundaries but might be worse at generating "spatially-smooth" predictions. Two sources of predictions complement each other, as we qualitatively illustrated in Figure 2 of the manuscript. Thanks to the proposed novel fusion strategy (Eq. 4), fusing SGC and decoder predictions leads to well-calibrated (evidenced by the ECE score in Fig. 5(a)) pseudo labels that can well utilize information of two sources, including the boundary pixels. Based on our empirical studies, we conjecture that Well-Calibrated (as evidenced by lower confidence-calibration metric - ECE score - in Fig.5 (a)) soft pseudo labels on the boundaries are vital and can still provide useful supervision signals for our consistency training framework. In contrast, self-training generates hard pseudo labels offline by confidence thresholding, where pixels with correct but not confident (likely boundaries) predictions will be discarded.
>
> **- Experiments on urban datasets.** We conducted additional experiments on the Cityscapes dataset. We trained both CutMix [A5] and our method using DeepLabv3+ with ResNet-101 backbone on the 1/4, 1/8, and 1/30 splits provided by CutMix [A5]. The comparison is shown as follows. Our method demonstrates clear performance improvement with larger margins in the lower data regimes. Please note that we use the exact same hyperparameters as other experiments, suggesting the insensitivity of our method to hyperparameters. Considering the high layout difference of Cityscapes to VOC/COCO, we believe there is further room for improvement with better hyperparameters. We will release the code to reproduce the experiments.
>
> |        | 1/4 (744) | 1/8 (372) | 1/30 (100) |
> |--------|-----------|-----------|------------|
> | CutMix | 68.33     | 65.82     | 55.71      |
> | Ours   | 72.36     | 69.81     | 60.96      |

---

> > ### Author Response · Authors · 2020-11-23
> > **Response to R3 - 2**
> >
> > **- Long-tailed class distribution of urban datasets.** We think the reviewer makes a great point of view on this challenge. We agree that the long-tail problem is a very challenging issue of SSL. It is an open question even for classification. Most advanced low-data regime SSL classification methods assume balanced labeled and unlabeled datasets. A recent work [A4] starts to address this challenge. Segmentation is likely more challenging. Our work does not, in particular, address this challenge. Currently, we can only provide some observation results to help better understanding. For example, on the supplementary cityscapes results with 1/30 splits, we find our method does not have a noticeable drop of performance on tail classes. For example, in the following table, we list the top-5 tail classes and their per-class mIoU of supervised baseline and PseudoSeg (class pixel ratios are shown brackets). PseudoSeg shows reasonable improvement over these classes. Additionally, since solving the long-tailed class distribution problem is an independent research problem, we will leave this study as future work. Thanks for the question, and it helps a deeper understanding of the algorithm.
> >
> > | mIoU (%) | motorcycle (0.06%) | bus (0.13%) | traffic light (0.14%) | rider (0.16%) | train (0.23%) |
> > |----------|--------------------|-------------|-----------------------|---------------|---------------|
> > | Baseline | 25.86              | 14.19       | 50.09                 | 43.91         | 17.78         |
> > | Ours     | 43.96              | 26.30       | 58.19                 | 46.09         | 22.13         |
> >
> > Reference
> >
> > [A1] Berthelot et al. “MixMatch: A Holistic Approach to Semi-Supervised Learning” NeurIPS 2019
> >
> > [A2] Grandvalet and Bengio. “Semi-supervised Learning by Entropy Minimization” NeurIPS 2005
> >
> > [A3] Xie et al. “Unsupervised Data Augmentation for Consistency Training” NeurIPS 2020
> >
> > [A4] Huyal el al. “Class-Imbalanced Semi-Supervised Learning”
> >
> > [A5] French et al. “Semi-supervised semantic segmentation needs strong, varied perturbations” BMVC 2020

---

### Official Review · AnonReviewer2 · 2020-10-28
**A simple and effective method for semi-supervised semantic segmentation**

**Rating:** 8
**Confidence:** 5

**Review:**

Summary:
This paper focuses on the problem of semi-supervised semantic segmentation, where less pixel-level annotations are used to train the network. A new one-stage training framework is proposed to include the process of localization cue generation, pseudo label refinement and training of semantic segmentation. Inspire by recent success in the semi-supervised learning (SSL), a novel calibrated fusion strategy is proposed to incorporate the concept of consistency training with data augmentation into the framework. Experiments on PASCAL VOC and MSCOCO benchmarks validate the effectiveness of the proposed method.

Pro:
+ The proposed one-stage training framework is elegant compared with two stage methods in this area which include one step for pseudo-label generation and another step for refinement then semantic segmentation training.
+ The new designed calibrated fusion strategy well incorporate the concept of consistency training with data augmentation into the same framework.
+ Achieve a new state-of-the-art on both PASCAL VOC and MSCOCO benchmarks compared with recent semi-supervised semantic segmentation methods.

Questions:
- CCT (Ouali et al., 2020) includes the consistency training with perturbations which can be treated as a kind of data augmentation on features. I'm wondering if authors can provide some insights about why the proposed method can achieve better performance than CCT when they both include the consistency training and data augmentation in the designs.
- In table 3, I suggest to include the segmentation framework used by each method in the table. In early works, old version of deeplab is usually treated as the standard. I understand using deeplab v3 is a fair comparison with CCT. It would be good to make this information clear in the table.
- It is also suggested to report the performance on PASCAL VOC test set as it is a common practice in this area (although CCT does not do so).
- Sine the unlabeled data training branch does not rely on any pixel-level annotations, I'm wondering if the proposed method can also work under weakly-supervised setting, where no pixel-level annotations are available during the training.

---

> ### Author Response · Authors · 2020-11-23
> **Response to R2**
>
> We thank R2 for appreciating our approach "elegant", "simple and effective".
>
> **1. Hypothesis of the comparison with CCT.**
>
> **(1) Augmentation:** CCT augments data in the *feature* space while we augment data in the *input* space. Considering the research development in the data augmentation field heavily towards input space augmentation [A4][A5][A6] and advanced SSL [A1][A2][A3] uses input space augmentation, our empirical experience conjectures that input augmentation is a more widely-developed choice than feature space augmentation.
>
> **(2) Ensemble view:** Both CCT and the proposed method can be interpreted from a model ensemble’s perspective [A7]. CCT generates different predictions via perturbing the internal feature and duplicating the network. As the network structure is the same, it may not be effective to generate diverse/distinct output predictions. On the other hand, our proposed method utilizes two distinct decision mechanisms to generate diverse outputs (as illustrated in Fig 2). We also validate that using the ensembled/fused output leads to better performance and well-calibrated pseudo labels (evidenced by a higher ECE score than (ensemble) alternatives in Figure 5(a)).
>
> **(3) Orthogonality:** Note that the contributions of CCT and that of the proposed method are *orthogonal*. Our method focuses on designing better pseudo labels and enables the utilization of stronger augmentation for consistency training.
>
> **2. Include the segmentation framework in Table3.** Thanks for your suggestion. We have updated Table 3 in the draft according to your feedback.
>
> **3. Report test set results on VOC.** Thanks for your suggestion. Due to the submission limitation from the official server (2 submissions for every seven days), we only report test set mIoU on the 1/16 split (compared with CutMix in fully-labeled+unlabeled setting and CCT in fully-labeled+image-level setting). We present the results below.
>
> | Pixel-level and unlabeled   |        |       |
> |-----------------------------|--------|-------|
> | Method                      | CutMix | Ours  |
> | mIoU (%)                    | 53.47  | 66.49 |
>
> | Pixel-level and image-level |        |       |
> |-----------------------------|--------|-------|
> | Method                      | CCT    | Ours  |
> | mIoU (%)                    | 56.68  | 58.69 |
>
> **4. Weakly-supervised learning setting?** The main focus of our work is on semi-supervised semantic segmentation. In our problem formulation, we assume that limited pixel-level labels are available. We utilize these annotations to supervise the SGC to refine the initial Grad-CAM score, one of the key novelties that lead to our efficient one-stage training framework. Using weakly-labeled or unlabeled data only is a scenario beyond our current scope. We leave it as future work.
>
> Reference
>
> [A1] Berthelot et al. “ReMixMatch: Semi-Supervised Learning with Distribution Alignment and Augmentation Anchoring” ICLR 2020
>
> [A2] Sohn et al. “FixMatch: Simplifying Semi-Supervised Learning with Consistency and Confidence” NeurIPS 2020
>
> [A3] Xie et al. “Unsupervised Data Augmentation for Consistency Training” NeurIPS 2020
>
> [A4] Zhang et al. “mixup: Beyond Empirical Risk Minimization”
>
> [A5] Yun et al. “CutMix: Regularization Strategy to Train Strong Classifiers with Localizable Features” ICCV 2019
>
> [A6] Cubuk et al. “RandAugment: Practical automated data augmentation with a reduced search space” CVPR Workshop 2020
>
> [A7] Guo et al. “On Calibration of Modern Neural Networks” ICML 2017

---

### Official Review · AnonReviewer5 · 2020-11-06
**An incremental approach showing good performances**

**Rating:** 6
**Confidence:** 4

**Review:**

**Summary:**

This paper introduces a model to improve semantic segmentation by using a limited amount of pixel-labeled data and unlabeled data or image-level labeled data. The authors use a Self-attention Grad-CAM (SGC) and segmenter to generate the pseudo-labels during training. The approach shows good results on Pascal VOC and COCO datasets and is well analyzed.


**Reasons for score:**

I do not think the technical contribution is strong enough for ICLR. The paper is incremental and the proposed approach is a combination of a lot of existing approaches. But I also want to highlight that the experimental section is strong and detailed.


**Pros:**

- The idea of using pseudo-labels is interesting because it allows to build larger dataset without increasing the annotation cost.
- The approach is evaluated in the settings of using unlabeled data and using image-level labeled data.
- The ablation study section gives a lot of details about the model. The authors analyzed a lot of things: expected calibration error, hypercolumn feature, soft vs hard label, temperature sharpening, color jittering strength, backbone architecture.
- The approach shows good results on Pascal VOC and COCO datasets.
- The proposed method achieves good performance in the low-data regime


**Cons:**
- The overall approach seems incremental because it is a combination of a lot of existing approaches and there is not a strong technical contribution. For instance, the model uses several loss functions and all the losses are jointly optimized.
- I think the related work section should be in the main paper instead of the supplementary.
- I feel some parts are a bit difficult to read because of some misleading information. For example, the title of section 3.1 is “Experiments using unlabeled data” but the model still uses some labeled data.

---

> ### Author Response · Authors · 2020-11-23
> **Response to R5**
>
> We thank R5 for the feedback and hope our response addresses the questions.
>
> **1. Contribution.** We respectfully disagree with R5 regarding the comments on the technical novelty of our work.
> Consistency-based training for SSL is an important problem in ML. Its recent success on image classification has not been fully demonstrated in dense prediction tasks such as semantic segmentation due to its particular challenge (e.g., designing high-quality structured pseudo labels for consistency training). Our work proposes an effective consistency training strategy without inducing additional training overhead as other two-stage algorithms [A5]. For specific technical contributions, our method not only presents a novel view of pseudo labeling strategy for semantic segmentation, but also proposes specific designs to solve this challenge using one-stage training. Overall, the targeted usage of known techniques is *significant beyond incremental*. We validate each component to demonstrate its necessity. We provided details as follows.
>
> **(1) SGC and calibrated fusion.** Please note Eq. 3 (self-attention for Grad-CAM score propagation, SGC) and Eq. 4 (calibrated fusion) are part of our contributions. Our results demonstrate their contributions to the final performance (e.g., 67.06% in fully-labeled+unlabeled data setting under the 1/16 split of VOC). We also compare with some relatively straightforward combinations of existing techniques:
> - Utilizing SGC (*53.42% in fully-labeled+unlabeled data setting under the 1/16 split of VOC*) or
> - Decoder (*53.20% in fully-labeled+unlabeled data setting under the 1/16 split of VOC*) as pseudo labels.
>
> These approaches follow similar designs of image classification methods [A1][A3]. Our results show that these baselines are less effective for semantic segmentation (in Table 5). Our proposed calibrated pseudo labels address the challenge of designing high-quality pseudo labels. R3 highlights this contribution.
>
> **(2) Novel utilization of existing techniques.** Our method does leverage some known and generic techniques. Nevertheless, each design has its clear rationale to solve a particular problem. These problems are different from those original techniques aimed at. For example
> - Hypercolumn features. The original purpose of hypercolumn features is to provide better features for object fine-grained localization. In our method, we leverage the hypercolumn features and show its importance in better capturing the semantic similarity between pixels in Figure 5(b).
> - Self-attention operation. The original self-attention dot-product is used to increase the receptive field or global context of CNN features. Here, we utilize self-attention for CAM score propagation and propose to use pixel-level labels to supervise it, **which is novel in the context of semantic segmentation**.
> - Calibrated fusion. We also introduce a novel fusion strategy (Eq. 4) and demonstrate its effectiveness from both performance (67.06% v.s. 53.20% in Table 5) and a calibration perspective (Figure 5(a)).
>
> **(3) Loss optimization.**  In terms of the usage of several losses, we argue that, in SSL, it is common to introduce auxiliary losses to leverage unlabeled data for supervision or regularization purposes, even in image classification [A1][A2][A4]. Our method for semantic segmentation in total has four loss terms and, more importantly, our design makes all losses just simple cross-entropy losses. Therefore, we respectfully argue that our proposed framework with four cross-entropy losses to solve this problem - instead demonstrates our method's advantage.
>
> **(4) Strong results.** The contribution of our work can also be evidenced by strong performance over different datasets, including the newly added experiments on the Cityscapes dataset (all achieved in a unique set of hyperparameters) as shown in response to R3, suggesting that our design leads to a stable and generalizable framework.
>
> **2. Related work and readability.** Thank you for your suggestion. We have updated the draft accordingly. The related work section is now Section 2 of the main paper.
>
> **3. Method exposition.** Thanks for your suggestions. We define the settings of "using unlabeled data" and "using image-labeled data" in the paper's introduction and method sections. We have revised the draft (mainly in the experimental section) to clarify further the setting to reduce confusion.
>
> Reference
>
> [A1] Berthelot et al. “ReMixMatch: Semi-Supervised Learning with Distribution Alignment and Augmentation Anchoring” ICLR 2020
>
> [A2] Lee. “Pseudo-Label:The Simple and Efficient Semi-Supervised Learning Method for Deep Neural Networks” ICML Workshop 2013
>
> [A3] Sohn et al. “FixMatch: Simplifying Semi-Supervised Learning with Consistency and Confidence” NeurIPS 2020
>
> [A4] Verma el al. "Interpolation Consistency Training for Semi-Supervised Learning", IJCAI 2019
>
> [A5] Ouali et al. “Semi-supervised semantic segmentation with cross-consistency training” CVPR 2020

---

### Public Comment · ~Wanxuan_Lu1 · 2020-11-13
**Some experiment setting is inconsistent with the previous semi-supervised segmentation methods.**

The basic settings of previous semi-supervised segmentation methods are: VOC and Cityscapes are used as experimental datasets; DeeplabV2 is the base model; The numbers of pixel-labeled images are 106, 211, 530, 1322.

1. The author combines grad-CAM to generate better pseudo-labels, but how to use grad-CAM in a dataset like Cityscapes where almost every image has the same image-level label?

2. In Table 2, the author makes a comparison with many semi-supervised segmentation methods, but most of these methods use DeeplabV2 as the base model. The full supervision's mIoU score of DeeplabV2 is lower at 3.5% than that of DeeplabV3+. It is better for authors to use the relative mIoU w.r.t full supervision when compared with previous methods.

3. In Table 2, the previous methods have not experimented on such numbers of pixel-labeled images (92, 183, 366, 732). Did the authors reimplement the previous method under this setting?

---

> ### Author Response · Authors · 2020-11-23
> **We have compared on both public splits and our splits to ensure a fair comparison.**
>
> Thanks for your comments.
>
> **1.** Our method is applicable to the Cityscapes dataset.
>
> First, although the scene layouts are quite similar for all the full images, it is still feasible to generate different image-level labels through a more aggressive geometric data augmentation (e.g., scaling, cropping, translation, etc.). In practice, standard segmentation preprocessing steps only crop a sub-region of the whole training images. It only contains partial images with a certain subset of image labels, making the training batches have diverse image-level labels (converted from pixel-level labels, in fully-labeled+unlabeled setting). Moreover, in the fully-labeled+weakly-labeled setting, in practice, we can collect diverse Internet images and weakly label them, instead of weakly labeling images from Cityscapes, which is a toy example setting. Please note that, at the inference stage, only decoder output is used.
>
> Second, we agree with the concerns that Grad-CAM is not optimal in this case. However, please note that our proposed SGC maps a refined version of raw Grad-CAM, which uses Grad-CAM as initial scores and has learnable parameters (supervised by pixel-level training images using $L_{sa}$ specified in Section 3.1 ) to refine it to better segment objects.
>
> As requested by R3, we conduct experiments on Cityscapes with the proposed method directly and find it achieves promising performance in the low-data regimes.
>
> |        | 1/4 (744) | 1/8 (372) | 1/30 (100) |
> |--------|-----------|-----------|------------|
> | CutMix | 68.33     | 65.82     | 55.71      |
> | Ours   | 72.36     | 69.81     | 60.96      |
>
> **2.** As R2 mentioned, a recent state-of-the-art CCT [A1] uses PSP-Net, which is comparable to our DeepLabv3+ model. We also include DeepLabv3+ results from another recent state-of-the-art CutMix segmentation paper [A2]. Thus, we are making fair comparisons here.
>
> **3.** We use the official released codes and follow their instructions to train these methods on the corresponding data splits. To further ensure a fair comparison, we also compare our method with others on the public 1.4k/9k split.
>
> Reference
>
> [A1] Ouali et al. “Semi-supervised semantic segmentation with cross-consistency training” CVPR 2020
>
> [A2] French et al. “Semi-supervised semantic segmentation needs strong, varied perturbations” BMVC 2020

---

### Author Response · Authors · 2020-11-23
**Summary of changes in the revised paper**

We thank the reviewers for their constructive feedback. According to the reviewers’ comments, we revise the paper in the following aspects (also highlighted in red in the paper):
1. We move the related work section from the appendix to the main paper (R5)
2. We further clarify our experiment setting in the result section (R5)
3. We include the segmentation network architecture in Table 3 (R2)
4. We conduct an additional study on distribution sharpening with different temperature values and include it in the appendix (R3)
5. We conduct additional experiments on Cityscapes and include the results in the appendix (R3)

---

### Decision · Program_Chairs · 2021-01-07
**Final Decision**

**Decision:**

Accept (Poster)

**Comment:**

The authors introduce an approach for designing pseudo-labels in semi-supervised segmentation.
The approach combines the idea a refining pseudo-labels with self-attention grad-CAM (SGC) and a calibrated prediction fusion, and consistency training by enforcing pseudo labels to be robust to strongly-augmented data.

The reviewers overall like idea and point out the good level of performance obtained by the method in the challenging semi-supervised context. However, they also point out the limited novelty of the approach, and the need for a better positioning with respect to related works. After rebuttal, reviewers were satisfied with authors' answers and paper modifications, and all recommend acceptance. \
The AC considers that the submission is a nice combination of existing techniques and likes the simplicity of the one-stage approach, which reaches good performances. Therefore, the AC recommends acceptance.